# Deep Learning Methods Applied to Drug Concentration Prediction of Olanzapine

**DOI:** 10.3390/pharmaceutics15041139

**Published:** 2023-04-04

**Authors:** Richard Khusial, Robert R. Bies, Ayman Akil

**Affiliations:** 1Department of Pharmaceutical Sciences, College of Pharmacy, Mercer University, Atlanta, GA 30341, USA; 2Department of Pharmaceutical Sciences, School of Pharmacy and Pharmaceutical Sciences, University at Buffalo, Buffalo, NY 14214, USA; 3Institute for Artificial Intelligence and Data Science, University at Buffalo, Buffalo, NY 14260, USA

**Keywords:** pharmacometrics, deep learning, population pharmacokinetics, drug concentration predictions, LSTM, neural networks, Bayesian optimization

## Abstract

Pharmacometrics and the utilization of population pharmacokinetics play an integral role in model-informed drug discovery and development (MIDD). Recently, there has been a growth in the application of deep learning approaches to aid in areas within MIDD. In this study, a deep learning model, LSTM-ANN, was developed to predict olanzapine drug concentrations from the CATIE study. A total of 1527 olanzapine drug concentrations from 523 individuals along with 11 patient-specific covariates were used in model development. The hyperparameters of the LSTM-ANN model were optimized through a Bayesian optimization algorithm. A population pharmacokinetic model using the NONMEM model was constructed as a reference to compare to the performance of the LSTM-ANN model. The RMSE of the LSTM-ANN model was 29.566 in the validation set, while the RMSE of the NONMEM model was 31.129. Permutation importance revealed that age, sex, and smoking were highly influential covariates in the LSTM-ANN model. The LSTM-ANN model showed potential in the application of drug concentration predictions as it was able to capture the relationships within a sparsely sampled pharmacokinetic dataset and perform comparably to the NONMEM model.

## 1. Introduction

Model-informed drug discovery and development (MIDD) is a valuable resource assisting in drug discovery and development, regulatory assessment, and life cycle management [1,2]. MIDD is an approach that aims to use models and integrate various data sources to generate information and knowledge to inform drug development and decision-making. Pharmacometrics and machine learning are approaches utilized within MIDD to aid in research and development projects. In particular, with regard to pharmacokinetic modeling, a popular approach toward the prediction of drug exposure (i.e., drug blood concentration predictions) has been population pharmacokinetic analysis utilizing nonlinear mixed effects (NLME) modeling. Although widely popular, NLME requires the generation of statistical models through an iterative trial-and-error process that can be time-consuming and labor-intensive [3,4,5,6,7].

There has been increasing interest in machine learning and deep learning approaches to accelerate and enhance the drug discovery and development process [8,9,10,11,12,13,14,15]. Several studies have been conducted to investigate the use of machine learning or deep learning approaches toward the prediction of drug blood or plasma concentrations. A backpropagation artificial neural network (BPANN) was constructed and validated to predict the plasma concentrations of rosuvastatin in healthy subjects [16]. The BPANN was also able to predict pharmacokinetic parameters with no significant differences from the measured pharmacokinetic parameters of rosuvastatin. Several machine learning algorithms (multiple learning regression, artificial neural networks, regression tree, multivariate adaptive regression splines, boosted regression tree, support vector regression, random forest regression, lasso regression, and Bayesian additive regression trees) were used for the prediction of a stable tacrolimus dose [17]. It was concluded that the regression tree performed the best among the machine learning algorithms in both the derivation and testing cohorts. A long short-term memory (LSTM) model was developed to assess its ability in predicting valproate concentrations in older patients with epilepsy [18]. When compared with the population pharmacometrics model of valproate, the developed LSTM model had a better predictive performance in the external evaluation study. Another study was conducted to see whether a gated recurrent unit (GRU) model could accurately predict the early induction kinetics of propofol in morbidly obese and lean subjects [19]. The generated GRU model with ensembled learning outperformed the compartmental pharmacokinetic model and had similar performances to the recirculatory model. Despite its potential and promise, many questions remain unanswered in terms of how deep learning can be used for the purpose of predicting drug blood or plasma concentrations. One area is the assessment of how deep learning algorithms can be tuned to generate more accurate predictions. Another area is how to generate explainable knowledge relevant to pharmacokinetics and drug development, such as subject-level covariates’ impacts on drug concentrations.

In deep learning, tuning models relies on the algorithm’s hyperparameters. Hyperparameters are critical to the behavior of deep learning algorithms. The computational time and memory cost to generate trained models, the quality of trained models, and the ability to accurately make predictions based on new inputs are some of the factors related to the hyperparameters of the model [20]. Bayesian optimization is a class of algorithms that specialize in optimizing hyperparameters in models that are expensive to evaluate, such as deep learning algorithms [21,22]. Bayesian optimization has shown success in optimizing deep learning hyperparameters in various applications [23,24,25,26,27]. It is certainly an attractive method to investigate for deep learning algorithms applied for the prediction of drug concentration predictions.

A shortcoming of deep learning algorithms is the limited explanation and interpretability of the information extracted from the selected features [28]. This is a hurdle in the drug development process because scientists and clinicians cannot assess the relevance and clinical impact of the features (i.e., covariates), deem a deep learning model scientifically appropriate, or the assess the reliability of results [29,30]. Therefore, the development of tools and methods that would provide an interpretation of deep learning models is critical. Permutation importance is a model-agnostic measure of feature importance. It assesses the intrinsic predictive value of a particular feature toward a model [31]. In the context of drug concentration prediction, permutation importance can assess the importance of different patient’s covariates toward model performance.

In this work, we present a novel deep learning framework for drug concentration prediction in a sparsely sampled pharmacokinetic study. An LSTM-ANN model with multiple inputs was constructed to predict olanzapine drug concentrations. Bayesian optimization with four different probabilistic surrogate models was used to optimize the LSTM-ANN hyperparameters. The performance of the optimized LSTM-ANN was compared to the performance of a NONMEM model. A measure of permutation importance was conducted to evaluate the importance of patient’s covariates toward the optimized LSTM-ANN performance. The novelty of the proposed layered design is that it provides the flexibility for multiple inputs. This provides the LSTM-ANN model a unique approach to fully maximize its ability to learn and extract distinctive patterns from different inputs. In addition, in the context of understanding the impact of patient-specific covariates on drug concentrations, we present a permutation analysis approach that was designed for the purpose of understanding covariate importance toward drug concentration predictions.

## 2. Materials and Methods

### 2.1. Study Population

Data used in this project were obtained from The National Institute of Mental Health Clinical Antipsychotic Trials of Intervention Effectiveness (CATIE) study. Access to the CATIE study data was provided by the NIMH data repository. The CATIE study consisted of two separate trials to determine the effectiveness of antipsychotics in patients suffering from either Alzheimer’s disease (AD) or schizophrenia (SZ). The in-depth rationale, study design, and methods have been published previously [32,33,34]. Briefly, both trials were conducted from October 2001 to December 2004 at 57 U.S. clinical sites. Inclusion and exclusion criteria of the CATIE study have been well documented in the literature [33,34,35]. A portion of patients from both trials was randomized to be given olanzapine, a second-generation antipsychotic. In the CATIE-AD trial, patients were given a dose of oral olanzapine from 2.5 to 20 mg/day taken once a day. In the CATIE-SZ trial, patients were given a dose of oral olanzapine from 7.5 to 30 mg/day taken once or twice a day. Plasma samples were collected during study visits along with records of the time of last dose and the time of sample taken. Patients provided from 1 to 6 plasma samples for olanzapine drug concentration analysis [36,37]. Demographic information of the olanzapine data from the CATIE-SZ and CATIE-AD studies were summarized using counts for both continuous and discrete covariates (Table 1).

### 2.2. Preprocessing of Data

For the population pharmacokinetic study, the data from the two trials were pooled together for a NONMEM analysis. For deep learning modeling, the data from the two trials were pooled together and then sorted by date in ascending order. The ordered pooled data was split into a training set (70% of data) and validation set (30% of data) for development and validation of the LSTM-ANN regression model. The MinMaxScaling function from the Scikit-learn Python package was used to transform all covariates and class variable within the range (0, 1) to aid in algorithm convergence. The equation used by MinMaxScaling was:(1)$$x_{\mathrm{scaled}}=\frac{x-x_{\min}}{x_{\max}-x_{\min}}$$
where x was a single observation of a feature. The $x_{min}$ was the minimum value of the feature, and $x_{max}$ was the maximum value of the feature. After scaling, the training set and validation set were separated based on the dosing variable, covariate, and the drug concentration (class) as shown in Figure 1.

### 2.3. Population Pharmacokinetics

Nonlinear mixed-effects modeling was performed using NONMEM 7.5 (ICON Software Development), and statistical analysis was performed in R version with package Xpose4. The typical population pharmacokinetics analysis includes the development of structural, statistical, and covariate models as shown in Figure 2.

A covariate analysis was conducted by evaluating the effect of subject-specific covariates on the model parameters. Continuous covariates were tested as follows:(2)$$\mathrm{TVCL}=\theta_{\mathrm{CL}}\times {\left(\frac{\mathrm{CC}}{{\mathrm{Median}}_{\mathrm{CC}}}\right)}^{\theta_{\mathrm{CC}}}$$
(3)$$\mathrm{CL}=\mathrm{TVCL}*\exp(\eta i_{\mathrm{CL}})$$
where TVCL was the typical value of clearance for the population; ηi was the random effect representing the difference between the ith subject and the population mean clearance. Random effects of BSV were assumed to be log distributed, with a mean of 0 and a standard deviation of ω. CC were the continuous covariates being tested against clearance.

Categorical covariates were tested as shown in the following example:(4)$$\mathrm{TVCL}=\left[\theta_{1}*\left(1-\mathrm{sex}\right)\right]+\left[\theta_{2}*\mathrm{sex}\right]$$
where sex=0 are females, and sex=1 are males. TVCL was the typical value of clearance for the populations. θ_1_ was the clearance estimate for females, and θ_2_ was the clearance estimate for males.

Both continuous and categorical covariates were tested in a stepwise fashion. A significance level of (*p* < 0.01) was used during the covariate search.

### 2.4. Neural Networks

All neural network model generation was conducted in Python version 3.9.7 with the TensorFlow 2.8.0 package. This study focused on two types of neural networks: artificial neural networks and long short-term memory networks. As shown in Figure 3a, a standard artificial neural network (ANN) architecture contains an input layer, one or more hidden layers, and one output layer. The input layer contains features to be entered into the model. The hidden layers learn and extract patterns within the data by nonlinear transformations by the way of activation functions. The output layer is the final layer of the network where the predictions are obtained.

Long short-term memory (LSTM) is a type of recurrent neural network designed to handle time series data. A standard LSTM contains a cell state, hidden state output, input gate, forget gate, and output gate. The architecture and equations that explain the behavior of an LSTM cell are shown in Figure 3b. The cell state ($C_{t}$) is the long-term memory of the cell, while the hidden state ($h_{t}$) is the short-term memory. The cell state transverses throughout all LSTM cells in a particular layer and is connected to the input and forget gate. The input gate ($i_{t}$) examines the input ($x_{t}$) and determines the amount to update the cell state. The forget gate ($f_{t}$) determines the amount of information from previous computations to be discarded within the cell state. The output gate ($o_{t}$) controls which parts of the cell state should be outputted at a particular time step [38]. Along with the states, the input gate, forget gate, and output gate work in unison to determine the amount of information to be extracted and processed for future nodes [39]. $U^{i},U^{f},U^{o},U^{g}$ are the weight matrix of the gates and their connection to the current input ($x^{t}$). $W^{i},W^{f},W^{o},W^{g}$ are the weight matrix of the gates and their connection to the previous hidden state [38].

The proposed LSTM-ANN model consisted of long short-term memory (LSTM) layers, dense layers, dropout layers, input layers, and a flatten layer. The structure of the LSTM-ANN model is presented in Figure 3c. The deep learning architecture incorporated multiple inputs to divide the pooled pharmacokinetic dataset into dosing variables and patient covariates. This was done to incorporate different algorithms and layers in separate stages of the deep learning workflow to maximize pattern extraction and model development. The dosing variables training set consisted of the Time Since First Dose (TSFD) and Dose. The dose variables training set was added to an input layer and fed to an LSTM layer. A dropout layer with a rate of 0.1 was added after the LSTM layer. The LSTM layer, followed by a dropout layer with a rate of 0.1, was denoted as L. After L, a flatten layer was added to complete the LSTM portion of the analysis. The covariate training set consisted of Age, Count (number of concomitant medications), Inducers (number of inducer concomitant medications), Inhibitors (number of inhibitor concomitant medications), African American Race, White Race, Other Race, Sex, Smoking, Substrate (number of substrate concomitant medications), and Weight. The covariate training set was added to an input layer. The flatten layer (last layer of the LSTM portion) was concatenated with the input layer that contained the covariate training data. This concatenated layer was fed to a dense layer. A dropout layer with a rate of 0.1 was added after the dense layer. The dense layer, followed by a dropout layer with a rate of 0.1, was denoted as A. After A, an output layer was added to complete the LSTM-ANN model.

### 2.5. Bayesian Hyperparameter Optimization

Bayesian optimization was used to optimize the hyperparameters of the LSTM-ANN model. Bayesian optimization utilizes past performance results to build a probabilistic model connecting hyperparameters to a probability score on the objective function. This probabilistic model is in the form of a surrogate model. Parameters of the Bayesian optimization algorithms can be found in Appendix A. The loss function set to minimize was the training mean square error (MSE).

The hyperparameters and their respective search spaces were the same for all four surrogate models and random search. Specifically, the hyperparameters to be tuned were the number of L (LSTM layer followed by a dropout layer with a rate of 0.1), number of LSTM nodes, number of A (dense layer followed by a dropout layer with a rate of 0.1), number of ANN nodes, learning rate, and number of epochs. The hyperparameters that were fixed for all four surrogate models and random search were the activation function for LSTM nodes, activation function for ANN nodes, optimizer function, batch size, and time steps. Table 2 presents the search space for each hyperparameters to be tuned and the option/value for the fixed hyperparameters.

The evaluation metric for the optimized LSTM-ANN model was validation root mean square error (RMSE). The generated model with the lowest validation RMSE from all the models was chosen as the final model. Bayesian hyperparameter optimization algorithms were implemented in Python version 3.9.7 with Python packages Optuna version 3.0.3 and Scikit-optimize version 0.9.0.

### 2.6. Permutation Analysis

To understand the importance of each covariate toward the optimized LSTM-ANN model, permutation importance (PIMP) was applied. Permutation importance is a model-agnostic feature importance metric that examines the importance of a feature toward a particular model [31]. To evaluate the importance of a particular covariate (C), its values are randomly shuffled to create a permutated covariate vector ($C_{p}$) and then entered into the optimized LSTM-ANN model. The difference between the permutated *RMSE*, ${RMSE}_{C_{p}}$, and the baseline unpermuted *RMSE* was measured, as shown below:(5)$${\mathrm{PIMP}}_{I}={\mathrm{RMSE}}_{C_{p}}\;-\;\mathrm{RMSE}$$
where *PIMP_I_* was the permutation importance of a particular covariate. The *RMSE_Cp_* and *RMSE* was the permutated *RMSE* and baseline unpermuted *RMSE*, respectively. This process was repeated 10 times to calculate the average and standard deviation *PIMP_I_* score. A higher *PIMP_I_* score indicated that the covariate C had a higher feature importance. This PIMP analysis was conducted in Python version 3.9.7.

## 3. Results

The pooled analysis dataset included 1527 olanzapine drug concentrations obtained from 523 patients. The CATIE-SZ study contributed 1327 olanzapine drug concentrations from 406 patients, while CATIE-AD study contributed 200 plasma olanzapine concentrations from 117 patients. The pooled analysis dataset had a higher percentage of male patients (63%) and higher percentage of white patients (66%). The median age of patients in the pooled analysis dataset was 45 years with an average weight of 84.43 kg. Patient demographics and characteristics are summarized in Table 1.

### 3.1. Population Pharmacokinetics

The structural model that best described the data in the pooled analysis dataset was a one-compartment model with linear absorption and elimination. The absorption rate constant Ka could not be reliably estimated, and it was therefore fixed to 0.5 h^−1^ based on a previously published population pharmacokinetic model [40]. Evaluation of statistical error models showed that an additive error model provided a good data fit. The estimated value of clearance was 15.90 L/h, and the estimated value of the volume of distribution was 2182 L.

Several patient-specific covariates were tested in a stepwise forward fashion. The results of the covariate analysis showed that several covariates had a significant effect on the model pharmacokinetic parameters. The development of the final model is summarized in Table 3. Three covariates (smoking, sex, and African American race) were added in a univariate forward selection. The final NONMEM model had a RMSE of 31.129. The observed concentrations vs. the individual predictions plot from the final model are shown in Figure 4a.

### 3.2. Neural Networks: Bayesian Hyperparameter Optimization

Four surrogate models and random search hyperparameter optimization algorithms were explored throughout training and validation of the LSTM-ANN model. Readings for each of the models can be found in Appendix A. The final model structure was optimized by Bayesian hyperparameter optimization with a TPE surrogate model with a hyperband pruner. The final model structure and training parameters are shown in Table 4. The best-performing model consisted of a single LSTM layer with eight nodes followed by two dense layers. The first dense layer had 88 nodes, and the second dense layer had 184 nodes. The optimal learning rate was 0.000125 with an ADAM optimizer. Figure 4 shows the relationship between the observed concentrations and individual predictions from NONMEM and the LSTM-ANN model. In both models, the observed concentrations correlated well with the individual predictions. In addition, both models had many concentrations near the identity line with a large density of concentrations within the range of 0–100 ng/mL. A plot on the log scale showing the relationship between the observed concentrations and individual predictions from NONMEM and the LSTM-ANN model can be found in Appendix A. The RMSE for the training set and validation set was 18.533 and 29.556, respectively.

### 3.3. Permutation Analysis

Permutation analysis was applied to the LSTM-ANN model of best performance based on Bayesian hyperparameter optimization with the TPE surrogate model and a hyperband pruner. Eleven covariates (age, sex, smoking, weight, African American race, white race, other race, substrate, inducers, inhibitors, and count) were shuffled 10-fold, and the average ${PIMP}_{I}$ was computed. A list was generated to rank the 11 covariates by the increase of ${PIMP}_{I}$ in the validation set, shown in Table 5. Age was the highest ranked covariate with an average ${PIMP}_{I}$ score of 4.733. Sex and smoking were the next highest-ranked covariates with average ${PIMP}_{I}$ scores of 3.403 and 2.283, respectively. The least influential covariates toward the optimized LSTM-ANN model were inhibitors and other race with average ${PIMP}_{I}$ scores of 0.158 and 0.147, respectively.

## 4. Discussion

The current approaches for drug concentration prediction within a population pharmacokinetic analysis are time-consuming and labor-intensive. In this work, we applied deep learning approaches to generate an LSTM-ANN model with multiple inputs to predict olanzapine drug concentrations from the CATIE study. Hyperparameter optimization of the LSTM-ANN model was achieved through Bayesian optimization with a tree-structured Parzen estimator (TPE) surrogate model and a hyperband pruner. The final optimized LSTM-ANN model had an RMSE of 29.556 in the validation set. Permutation analysis revealed age, sex, and smoking as influential patient covariates toward model performance.

Nonlinear mixed modeling was conducted in NONMEM to provide a benchmark model for olanzapine drug concentration prediction from the CATIE study. A one-compartment model with additive error was then selected as the structural and statistical models, respectively. Smoking, sex, and African American race were three patient covariates that were selected in a stepwise search and added to the base model to produce a final model. A similar analysis evaluating the magnitude and variability of concentration exposure of olanzapine from the CATIE study was previously published by Bigos et al. [41]. In that analysis, the final model was a one-compartment model with additive error including smoking, sex, and African American race as significant patient covariates. The RMSE of the final model was 31.129.

The main feature of the deep learning architecture is the use of multiple inputs to split the pooled pharmacokinetic dataset into dosing variables and patient covariates. This provided the LSTM-ANN model a unique approach to fully maximize its ability to learn and extract distinctive patterns. There was little to no change in patients’ covariates throughout the trials. Due to this lack of variation, the LSTM algorithm would not need to learn long-term dependencies within the patients’ covariate data. The separation between dosing variables and patient covariates had a reduction of noise of the time-dependent signal within the dosing variables as compared with a combined dataset. The extracted time-dependent pattern by the LSTM was concatenated with the patient covariates and fed into the ANN algorithm. Relationships within the concatenation of the time-dependent pattern from the dosing variables and patient covariates were enhanced due to the reduction of noise as opposed to an input of all variables simultaneously. This approach benefited the ANN algorithm’s ability to learn and extract distinct relationships within the patient covariates and the dosing variables.

Initially, dosing variables without the respected patient’s covariates were fed into the LSTM-ANN model. An LSTM layer with a time step of two was used to learn and extract the time-dependent patterns within dosing variables. Following the LSTM layer was a dropout layer with a rate of 0.1. Dropout layers reduce neuron codependency within a chosen layer by selecting neurons at random to ignore at a given rate. Lowering neuron codependency has shown success in decreasing model overfitting [42,43,44]. After the dropout layer, a flatten layer was applied to the workflow. Within a LSTM structure, there is a hidden state that reflexes the short-term memory within the cell. At a given output, an LSTM only uses the very last hidden state for pattern extraction. This limits the amount of information the LSTM cell could use for prediction. With the use of a flatten layer, the LSTM transforms from a matrix to a vector, allowing the use of all previous hidden states for pattern extraction. This ultimately leads to an increase in model performance due to the increase of information [45]. The time-dependent patterns learned from the dosing variables were concatenated with their respected patient covariates. This concatenated layer was used as an input layer for a standard ANN layer. After the patterns formed from the concatenation of dosing variable patterns and patient covariates were learned and extracted from the ANN layer, a dropout layer with a rate of 0.1 was added to limit model overfitting. Lastly, an output layer was required to generate prediction results.

Due to the complex nature of the deep learning architecture, Bayesian hyperparameter optimization algorithms were incorporated within the protocol. Each of the four surrogate models were able to achieve minimization of the training mean square error (MSE), shown in Appendix A. Tree-structured Parzen estimators (TPE) with a hyperband pruner reached minimization with the simplest model hyperparameter structure. Simpler models tend to be preferred due to their computational efficiency, increased interpretability, and prevention of overfitting [38,46]. The RMSE of the final optimized LSTM-ANN model was 29.556 in the validation set, which is lower than the RMSE obtained from the NONMEM model. The NONMEM model utilized the entire pooled pharmacokinetic dataset, while the LSTM-ANN model only used the training data to generate a prediction model. Both models were able to have similar success in prediction from the 0 to 50 ng/mL range; however, the NONMEM model had better success in predicting concentrations in the range of 50–150 ng/mL while the LSTM-ANN underperformed. Upon closer inspection of the training data, there were only 251 olanzapine concentrations within the range of 50–150 ng/mL, which accounted for only 23.5% of the total training data. Due to the low number of concentration samples, we believe the LSTM-ANN model struggled to learn the patterns and relationships within this range for prediction in the validation set.

To evaluate the impact of the patient specific covariates included in the analysis dataset on the LSTM-ANN model, a permutation analysis was conducted. To ensure reliable and robust results, each patient covariate was shuffled 10-fold, and the average of the results were computed. Age, sex, and smoking were three patient covariates that displayed a large impact on model performance. Sex and smoking were two patient-specific covariates that were found to be both statistically significant in the NONMEM covariate selection analysis and impactful in the performance of the LSTN-ANN model. This is in accordance with previous literature findings examining significant factors in olanzapine dosing [47,48,49,50]. However, to a lesser extent, the remaining patient covariates have shown to have an impact on model performance. We believe the LSTM-ANN model was able to utilize other patient covariates to learn different patterns and relationships to aid in model performance.

There are several advantages to our approach toward drug concentration prediction. The LSTM-ANN model requires no a priori assumptions on the nature of the relationships within the analysis data. The model is data driven without a specific model structure defined beforehand and without bounds of statistical assumptions. The LSTM-ANN model requires no patient covariate selection; instead, it explores and learns the relationships within all the patient covariates. With the increase of information, it is highly probable that the model’s performance will improve, leading to an overall result of more accurate drug concentration predictions. Another advantage is the unique LSTM-ANN architecture. With little time-specific variation within the patient covariates, we decided to separate the covariates from the dosing variables. With the use of multiple inputs, we used LSTM to learn time-specific patterns within the dosing variables. These patterns were extracted and added to their respected patient specific covariates. To learn the new relationships between the patterns extracted from the LSTM and their respected patient specific covariates, ANN layers were utilized. We believe the structure of this LSTM-ANN model aids in noise reduction, allowing important patterns within the data to be highlighted.

Despite the promising results, a few unanswered questions remain. The LSTM-ANN model’s ability to learn with varying sample sizes is an area that needs exploration. Typically, a larger sample size will aid in deep learning performance; however, smaller datasets may benefit from the reduction of noise in the beginning steps of the LSTM-ANN design. The model’s construction and validation were conducted in the same population. Further studies are necessary to explore our LSTM-ANN model’s generalizability. Last, an area that needs further studies for any deep learning based pharmacokinetic model is model interpretability. Explainable AI is an area of machine learning currently working on approaches to assist in translating deep learning results. Novel techniques from this field may aid in interpreting the results from the LSTM-ANN model.

Extrapolation outside the training range is an obstacle for any data-driven model, and our results are consistent with previous findings from other deep learning approaches toward drug concentration prediction. Previously, Liu et al. explored the application of LSTMs toward pharmacokinetic-pharmacodynamic modeling with varying dosing regimens. It was shown that the developed LSTM model was able to reasonably predict the pharmacodynamics profile under the QD regimen; however, it struggled to predict well with BID or TID extrapolation [51]. In another study, Lu et al. investigated machine learning and deep learning approaches toward drug concentration predictions in unseen dosing schedules and compared them with traditional NLME modeling. It was demonstrated that machine and deep learning approaches did not extrapolate well, and a new deep learning approach, Neural-ODE, was introduced [8]. Our findings are consistent with the published literature showing that the LSTM-ANN model did not perform as well in predicting drug concentration that were not well represented in the training dataset. It is worth noting that the CATIE study was designed to be a sparsely sampled study for pharmacokinetic analysis. This, in turn, means that there are large gaps of time left for the LSTM-ANN to fill. Further studies exploring how such deep learning models can be optimized to handle sparsely sampled pharmacokinetics datasets are warranted.

## 5. Conclusions

In conclusion, an LSTM-ANN model was successfully developed to analyze olanzapine pharmacokinetics and predict drug plasma concentrations. The model produced similar results to that of a NONMEM model developed using the same dataset. A permutation analysis for covariates impact evaluation applied on the LSTM-ANN model produced similar results to that of stepwise covariate analysis in NONMEM. This work shows the potential of, and adds to the growing evidence to the role of, deep learning approaches in the field of pharmacometrics and model-informed drug development.

## Figures and Tables

**Figure 1 pharmaceutics-15-01139-f001:**
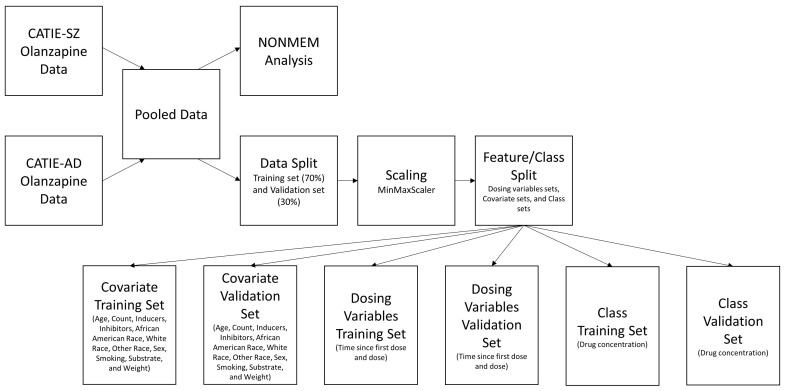
The proposed workflow began with pooling data from the two CATIE trials. The pooled data were used for both the NONMEM analysis and the LSTM-ANN model. The LSTM-ANN model data preprocessing consisted of a data split, followed by scaling, and dividing the data into multiple input features and class training and validation sets.

**Figure 2 pharmaceutics-15-01139-f002:**
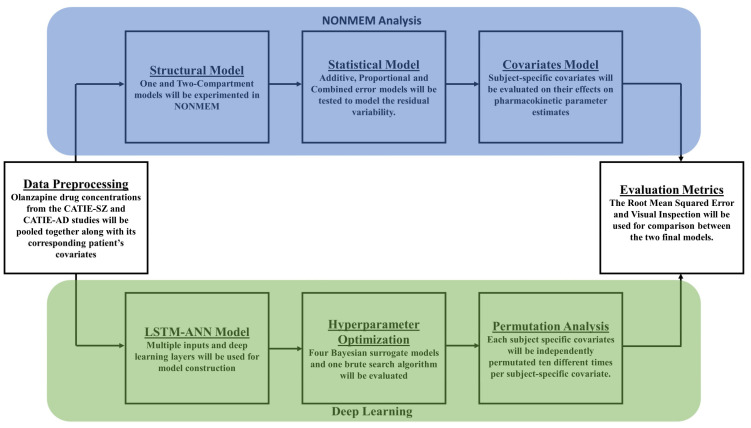
A flowchart showing model development for both the NONMEM analysis and the Deep Learning approach. The different colors correspond to the different modeling workflows. The blue represents the NONMEM analysis. Initially, a structural model was built, followed by a statistical model. Finally, a covariate analysis was completed. The green represents the Deep Learning approach. First, the LSTM-ANN model was built, followed by a hyperparameter optimization. Last, a permutation analysis was conducted.

**Figure 3 pharmaceutics-15-01139-f003:**
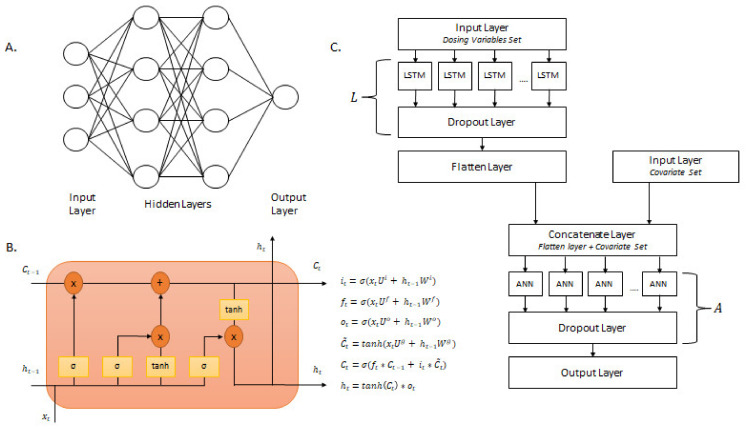
Model architecture of Deep Learning models. (**A**) Structure of an artificial neural network with an input layer, hidden layer, and an output layer; (**B**) Structure of the LSTM cell and the equations that refer to the gates of the LSTM cell; (**C**) Schematic of the LSTM-ANN model. Each box represents a different layer within the Deep Learning architecture.

**Figure 4 pharmaceutics-15-01139-f004:**
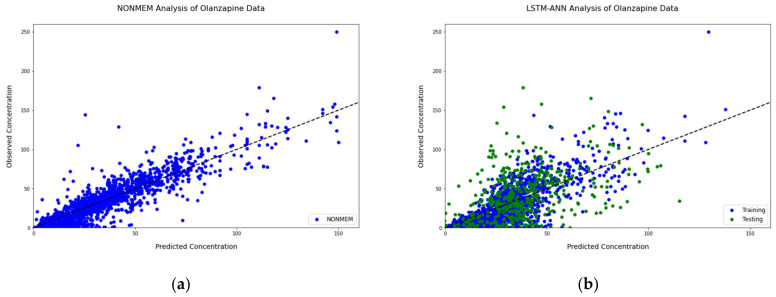
Observations vs. Predictions plot. (**a**) NONMEM (note: predictions are individual predictions (IPRED]); (**b**) LSTM-ANN (The different colors correspond to drug concentration predictions from the different datasets. Blue represents the training data, and the green represents the validation data).

**Table 1 pharmaceutics-15-01139-t001:** Demographic information of each study and pooled population.

	All Patients (*n* = 523)	CATIE-SZ (Schizophrenia Study) (*n* = 406)	CATIE-AD (Alzheimer’s Disease Study) (*n* = 117)
Observations	1527	1327	200
Age, median years ± SD (range)	45 ± 18 (18–103)	42 ± 10.9 (18–65)	78 ± 8.5 (45–103)
Race, (*n*) White Black/African American Asian American Indian Two or more races			
	346	253	93
	149	131	18
	19	14	5
	5	4	1
	4	4	0
Sex, (*n*)			
Male	332	289	43
Female	191	117	74
Smoking, (*n*)			
Active Smoker	274	267	7
Nonsmoker	249	139	110
Weight, mean weight (kg) ± SD	84.43 ± 22.1	89.34 ± 21.4	67.42 ± 15.07

**Table 2 pharmaceutics-15-01139-t002:** Hyperparameters that will be fixed and tested in the LSTM-ANN model.

Hyperparameters of the LSTM-ANN Model	
Hyperparameters to Be Tuned	Range to Be Tested
Number of L (LSTM + Dropout)	1–3 layers
Number of LSTM nodes	8–256 nodes
Number of A (Dense + Dropout)	1–3 layers
Number of ANN nodes	8–256 nodes
Learning Rate	0.001–0.0001
Number of Epochs	40–120 epochs
Hyperparameters to stay constant	Fixed Option/Value
Activation function for LSTM nodes	ReLU
Activation function for ANN nodes	ReLU
Optimizer function	ADAM
Batch Size	1
Time Steps	2

Abbreviations: ANN—Artificial Neural Networks, LSTM—Long Short-Term Memory, ReLU—Rectified linear unit.

**Table 3 pharmaceutics-15-01139-t003:** Changes in objective function with the addition of influential covariates.

Model	Objective Function	Decrease in Objective Function	
		From Base Model	From Previous Model
Base Model (Structural and Statistical Model)	10,419.333	N/A	N/A
Base Model + Smoking Status	10,374.054	45.279	45.279
Base Model + Smoking Status + Sex	10,361.536	57.794	12.518
Base Model + Smoking Status + Sex + Black/African American Race	10,352.008	67.325	9.528

Abbreviations: N/A—Not applicable.

**Table 4 pharmaceutics-15-01139-t004:** Optimized hyperparameters for the final LSTM-ANN model structure.

Optimized Final Model Structure	
Hyperparameters	Option/Value
Time Steps	2
Number of L (LSTM + Dropout)	1 layer
Number of LSTM nodes	8 nodes
Activation function for LSTM nodes	ReLU
Number of A (Dense + Dropout)	2 layers
Number of ANN nodes in Layer 1	88 nodes
Number of ANN nodes in Layer 2	184 nodes
Activation function for ANN nodes	ReLU
Optimizer function	ADAM
Learning rate	0.000125
Number of Epochs	69 epochs
Batch Size	1 batch

Abbreviations: ANN—Artificial Neural Networks, LSTM—Long Short-Term Memory, ReLU—Rectified linear unit.

**Table 5 pharmaceutics-15-01139-t005:** Mean PIMP scores from the permutation analysis of the LSTM-ANN model.

Permutation Analysis toward Covariate Importance	
Covariates	Weight (Average ± SD)
Age	4.733 ± 0.461
Sex	3.403 ± 0.683
Smoking	2.283 ± 0.399
White Race	1.936 ± 0.484
Weight	1.427 ± 0.374
Substrate	1.338 ± 0.415
Black/African American Race	1.204 ± 0.474
Count	0.844 ± 0.436
Inducers	0.730 ± 0.285
Inhibitors	0.158 ± 0.316
Other Race	0.147 ± 0.177

## Data Availability

The CATIE-SZ and CATIE-AD studies are available on request from the National Institute of Mental Health (NIMH). The CATIE-SZ data can be found here: [https://nda.nih.gov/edit_collection.html?id=2081]. The CATIE-AD data can be found here: [https://nda.nih.gov/edit_collection.html?id=2159]. Both the CATIE-SZ and CATIE-AD was accessed on 23 October 2020.

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
