# Peer review of "Deep Learning Methods Applied to Drug Concentration Prediction of Olanzapine"

_pharmaceutics, 2023, doi:10.3390/pharmaceutics15041139_

Round 1
Reviewer 1 Report
Dear Sir,
The work titled "Deep Learning Methods applied to Drug Concentration Prediction of Olanzapine" study focus on a deep learning model, LSTM-ANN to predict olanzapine drug concentrations from the CATIE study using a large number of sample and 11 patients.
The comments are as follows:
1. Why LSTM-ANN model preferred over BISTM-ANN?
2. Please discuss the comparison of the LSTM-ANN and BISTM-ANN models as well as any other recently developed one for similar work.
3. The data used is why so old, October 2001 through December 2004?
4. Figure 1 Workflow is originally developed or modified from any previous work, if modified please cite it.
5. Need to improve the quality of Figure 2 along with content, too much text. reduce it or present it in table form.
6. Figure 2 Workflow is originally developed or modified from any previous work, if modified please cite it.
7. Is the LSTM-ANN model validated before use?
8. Line 174: What author mean by, A dropout layer with rate 0.1 was added after the LSTM layer.
9. Figure 3 Workflow is originally developed or modified from any previous work, if modified please cite it.
10. Line 195-196, Editor can final call of this text, Parameters of the Bayesian optimization algorithms are shown in supplemental table #1.
Will it be appropriate to use a callout of the table that will not display after publication?
11. Table 2: In what basis Range to be tested, decided?
12. Line: 254-255, Readings for each of the models are given in supplementary table 2. The editor can take a final call.
Will it be appropriate to use a callout of the table that will not display after publication?
13. Improve Figure 4, legends, increase size and remove figure caption from an image as its duplicating.
14. Please modify the reference style as per MDPI guidelines.
Reviewer 2 Report
The authors performed a nice study regarding deep Learning methods applied to drug concentration Prediction of Olanzapine. However, the author needs to address the following comments;
1. The introduction part should be improved. The previous studies and the novelty of this research should be involved.
2. Detailed information about the inclusion and exclusion criteria of the selected criteria should be added.
3. Why did the author choose this study although it was performed many years ago?
4. Please add abbreviations under each table.
5. Please add conclusion as a separate part.

Reviewer 3 Report
The authors described a machine-learning method that would allow the prediction of olanzapine concentrations in a clinical setting. They compared the predictive performance of their model with the classical, population-based pharmacokinetic analysis. They found some pitfalls in the algorithm; despite thorough training, the model did not extrapolate well and failed to capture some outliers that were not present in the training set. The analysis was challenging because it included sparsely-sampled data.
The Reviewer has several comments that the authors should address:
1. The citations need to be more accurate. For example, reference 39 in line 239 should point to the paper on the popPK model for olanzapine. Yet, it is a paper on neural networks. All of the references have to be re-checked before the final submission.
2. Why did the authors not assess the influence of the study arm and diagnosis? There may be some disease-related issues that impact drug PK. Either way, the authors should justify why they did/did not account for it.
3. Why was the whole dataset used for building the popPK model? With such a large number of subjects and samples, the authors could break the group into the training and validation subset. Then, in the first stage, the initial model is built and tested against the validation set. If the predictions are acceptable, they could merge both sets and recalculate the estimates.
4. While building the popPK model, the forward-inclusion-backward-elimination is usually applied for covariate selection. In the present work, the authors limited the study to the forward-inclusion. Could the authors explain such a strategy?
5. The authors seem to assess the "comparability" of NONMEM and ML predictions based on RMSE. The question is - how was RMSE calculated? Often, especially in the field of popPK, the residual error is weighted or normalized for the observation, and a relative error is used for assessing the deviations from the true results. For example, suppose the actual results are 100 ng/mL and 1 ng/mL. The predictions are 105 ng/mL and 6 ng/mL, respectively. Both predictions deviate by 5 ng/mL, but the relative error is much greater for the lower predictions. The Reviewer thinks the better RMSE of the ML model (18.533 and 29.556 vs. 31.129) was caused by a much better fit for the observations in the 0 - 50 ng/mL range. However, as seen in Figure 4, the NONMEM model's points are less spread in the middle part of the graph and more evenly dispersed along the identity line. Speaking of Figure 4, please, rescale the left graph so the y-axes match in both panels. Also, please consider presenting it on a log-log scale. It will show the low concentrations much better.
6. After reading the manuscript, the Reviewer believes that the conclusion that the LSTM-ANN model performs comparably with the NONMEM model is too general. The authors could focus on where the similarities are and what differentiates one model from another (for example, in terms of selected covariates or the concentration range where the model performed best).
Round 2
Reviewer 3 Report
The Authors adequately responded to all the Reviewer's comments. No further elaborations are necessary.